# Polycomb and Notch signaling regulate cell proliferation potential during *Caenorhabditis elegans* life cycle

Francesca Coraggio[1,2,*], Ringo Püschel[1,2,*], Alisha Marti[1], Peter Meister[1]

**Stable cell fate is an essential feature for multicellular organisms in which individual cells achieve specialized functions. *Caenorhabditis elegans* is a great model to analyze the determinants of cell fate stability because of its invariant lineage. We present a tractable cell fate challenge system that uses the induction of fate-specifying transcription factors. We show that wild-type differentiated animals are highly resistant to fate challenge. Removal of heterochromatin marks showed marked differences: the absence of histone 3 lysine 9 methylation (H3K9) has no effect on fate stability, whereas Polycomb homolog *mes-2* mutants lacking H3K27 methylation terminally arrest larval development upon fate challenge. Unexpectedly, the arrest correlated with widespread cell proliferation rather than transdifferentiation. Using a candidate RNAi larval arrest-rescue screen, we show that the LIN-12[Notch] pathway is essential for hyperplasia induction. Moreover, Notch signaling appears downstream of food-sensing pathways, as dauers and first larval stage diapause animals are resistant to fate challenge. Our results demonstrate an equilibrium between proliferation and differentiation regulated by Polycomb and Notch signaling in the soma during the nematode life cycle.**

## Introduction

During development, the differentiation potential of cells is progressively restricted, and differentiated cells have mostly lost their plasticity. *Caenorhabditis elegans* conforms to this paradigm: early embryonic blastomeres can be converted into a number of cell types by ectopically expressing "selector" transcription factors (Horner et al, 1998; Zhu et al, 1998; Gilleard & McGhee, 2001; Quintin et al, 2001; Fukushige & Krause, 2005), whereas later during development, most cells lose this capacity. In fully differentiated animals, a single transcription factor, the endodermal-specifying ELT-7 is able to induce transdifferentiation of pharyngeal cells into an intestinal cell–like cell type (Riddle et al, 2013). Nematodes

are an interesting system to characterize the molecular players modulating somatic cell fate–plasticity during development (Hajduskova et al, 2012). Previous studies showed that in embryos, the elimination of the Polycomb complex or GLP-1[Notch] signaling extends the plasticity period of the blastomeres (Yuzyuk et al, 2009; Djabrayan et al, 2012). In the germline, chromatin remodelers and the Polycomb complex, repress plasticity and impair direct reprogramming into neurons (Tursun et al, 2011; Patel et al, 2012; Kolundzic et al, 2018). In contrast, GLP-1[Notch] signaling enhances transcription factor–induced cell plasticity, apparently independently of its proliferation-inducing function (Seelk et al, 2016). In differentiated animals, only a few factors are known to modulate cell plasticity, most of which were characterized in a natural transdifferentiation event, the endodermal Y to neuronal PDA conversion (Richard et al, 2011; Kagias et al, 2012; Zuryn et al, 2014; Kolundzic et al, 2018). Chromatin modifications appear to play a prominent role, as the temporally controlled expression of distinct histone modifiers is necessary for conversion (Zuryn et al, 2014). Here, we report a single-copy cell fate–induction system for the muscle and endoderm. Using muscle induction, we show that cell fate is remarkably stable in fully differentiated animals of the first larval stage as only one cell is able to transiently express muscle markers. In contrast, in the absence of the Polycomb complex, muscle fate induction leads to a robust developmental arrest and the presence of additional cells expressing the muscle marker. Using the invariant lineage of the nematode and cell type–specific fluorescent reporters, we show that these cells unexpectedly do not originate from a transdifferentiation event, but from re-entry into the cell cycle of normally terminally differentiated muscle cells. In addition, a number of other lineages including the neuronal ventral cord progenitors P, the mesodermal founder M, and the seam cell lineage V divide. For the seam cell lineage V, this occurs in the absence of previous DNA replication, leading to mitotic catastrophe and arrested anaphases, presumably leading to a nonfunctional hypoderm and developmental arrest. To understand how cell fate challenge can induce cell cycle entry, we carried out a candidate RNAi screen. We show that knock-down of the Notch signaling pathway can rescue both the developmental arrest upon cell fate

[1]Cell Fate and Nuclear Organization, Institute of Cell Biology, University of Bern, Bern, Switzerland   [2]Graduate School for Cellular and Biomedical Sciences, University of Bern, Bern, Switzerland

Correspondence: peter.meister@izb.unibe.ch
*Francesca Coraggio and Ringo Püschel contributed equally to this work

challenge and the cell cycle defects of Polycomb mutants. Accordingly, ectopic expression of muscle-inducing transcription factors led to increased expression of LAG-2, the single Notch ligand in *C. elegans*. As Notch signaling was previously shown to regulate entry and exit into the resistance stages of the nematode life cycle, we explored whether dauer and first larval stage animals in diapause are sensitive to cell fate challenge. We find that these animals are completely resistant to transcription-factor–induced transdifferentiation; hence, cell plasticity is highly reduced during these stages. Our results demonstrate a life-cycle stage–dependent regulated balance between Notch signaling–inducing cell proliferation and H3K27 methylation–protecting cells against unscheduled cell proliferation in the soma of *C. elegans*.

# Results

### A single-copy insertion system for cell fate stability challenge

To probe cell fate plasticity, we integrated single copies of a heat-shock (HS) inducible construct driving either a muscle (Fukushige & Krause, 2005) (*hlh-1*/MyoD) or an endoderm (Zhu et al, 1998) (*end-1*/GATA1) specifying transcription factor. Expression of the transcription factor is assessed by the red fluorescence from a trans-spliced *mCherry* ORF placed downstream of the transcription factor (Fig 1A). Muscle cells are identified by the expression of *gfp::histone* H2B under the transcriptional control of the heavy-chain myosin promoter *myo-3*. The expression of the tissue-specific inducers challenges cell fate (Fig 1A) (Yuzyuk et al, 2009): cells with a stable fate remain insensitive to the induction, whereas unstable ones will possibly transdifferentiate and express the terminal cell fate marker (here *myo-3::H2B*). As previously shown with multicopy array systems (Zhu et al, 1998; Fukushige & Krause, 2005), HLH-1[ect.] or END-1[ect.] expression in early embryos (20–100 cells) led to irreversible developmental arrest (Fig 1B). In addition, HLH-1[ect.] expression led to cellular twitching about 10 h post-induction and a significant number of cells expressed the nuclear muscle marker, suggesting transdifferentiation into muscles (Fig 1B, HLH-1[ect.], arrows, Fig S1A). As previously reported (Yuzyuk et al, 2009), cellular plasticity is lost later during embryonic development and expression of either transcription factor had no phenotypic effect and animals hatched normally after induction at the 16E stage (Fig S1B, WT).

### Cell fate of differentiated animals is robust

The fluorescent muscle fate reporter allows the visualization of potentially transdifferentiating cells beyond embryonic development. When HLH-1[ect.] was expressed in fully differentiated first larval stage animals, no obvious phenotypic defects were observed (Fig 1C). However, 24 h post-induction (animals in L2-L3 stage), about half of the animals had one additional cell than the normal 96 cells expressing the muscle marker (Fig 1D). This additional cell was located in the tail region and is present in 46% of the animals (n = 122), but never observed in control HS larvae (n = 91). This location corresponds to the anal sphincter cell (Sulston & Horvitz,

1977) (Fig 1E, arrow). Indeed, a cytoplasmic *myo-3p::RFP* marker highlighted the typical, saddle-like shape of this cell in 9 of 13 worms 24 h post–*hlh-1* induction (Fig 1E, bottom right). 48 h post-induction, expression of the muscle marker in the anal sphincter cells was no longer visible (Fig S2). We conclude that upon HLH-1[ect.] expression, the anal sphincter cell transiently expresses muscle-specific markers but subsequently represses these, supposedly reverting to its normal fate. Interestingly, this cell is lineally close to a muscle as its sister cell is a body wall muscle (Sulston & Horvitz, 1977; Fox et al, 2007). Reversion to silencing of the muscle marker might be a consequence of signaling from surrounding cells inhibiting complete fate conversion (Gurdon, 1988). Altogether, our experiments demonstrate that cell fate in differentiated animals is resistant to induction of muscle transdifferentiation.

### Absence of H3K9 methylation or perinuclear H3K9me anchoring has no effect on cell plasticity in differentiated animals

H3K9-methylated heterochromatin formation and its anchoring at the nuclear periphery has been shown to help stabilize ectopically-induced cell fates in embryos (Gonzalez-Sandoval et al, 2015). We therefore asked whether this feature could be extended to the first larval stage by testing mutants deficient for either H3K9 methylation or perinuclear anchoring of methylated H3K9 (*set-25 met-2* or *cec-4* mutants, respectively) (Towbin et al, 2012; Gonzalez-Sandoval et al, 2015). In both mutants, expression of HLH-1[ect.] did not lead to obvious phenotypic alterations, nor did it increase the total number of cells expressing the muscle marker or the proportion of animals in which the anal sphincter cell was positive for the *myo-3* marker (data not shown and Fig S3). As for wild-type animals, muscle marker expression was no longer observed in this cell 48 h post-induction. In conclusion, ablation of H3K9 methylation or its perinuclear anchoring does not affect cellular plasticity in fully differentiated animals.

### Absence of the Polycomb repressive complex leads to larval arrest upon cell fate challenge

H3K27 methylation, deposited by the Polycomb repressive complex 2, plays a crucial role in the modification of the epigenetic landscape during development (Bender et al, 2004; Niwa, 2007; Schuettengruber et al, 2007). Ablation of PRC2 components leads to an elongation of the embryonic plasticity window and renders germline cells amenable to fate conversions (Yuzyuk et al, 2009; Patel et al, 2012) (Fig S1B). Progeny of animals treated with *mes-2* (*RNAi*) and *mes-2* homozygous mutant animals develop normally, but are sterile in the second generation. This generation has no detectable H3K27 methylation; hence, the mark is dispensable for cell fate specification under normal growth conditions (Holdeman et al, 1998; Gaydos et al, 2012). Ectopic expression of HLH-1 in the first larval stage of second generation *mes-2* homozygous mutant animals has dramatic effects: 93% of the worms arrest larval development (Fig 2B and D, compared with 2A). The germline cell number confirms that most animals arrest at the L1 stage (4 Z cells, Fig 2B insert). Developmental arrest is a consequence of HLH-1[ect.] expression and not of the HS used for *hlh-1* induction as heat-treated *mes-2* animals develop normally (Fig 2A). Although motile

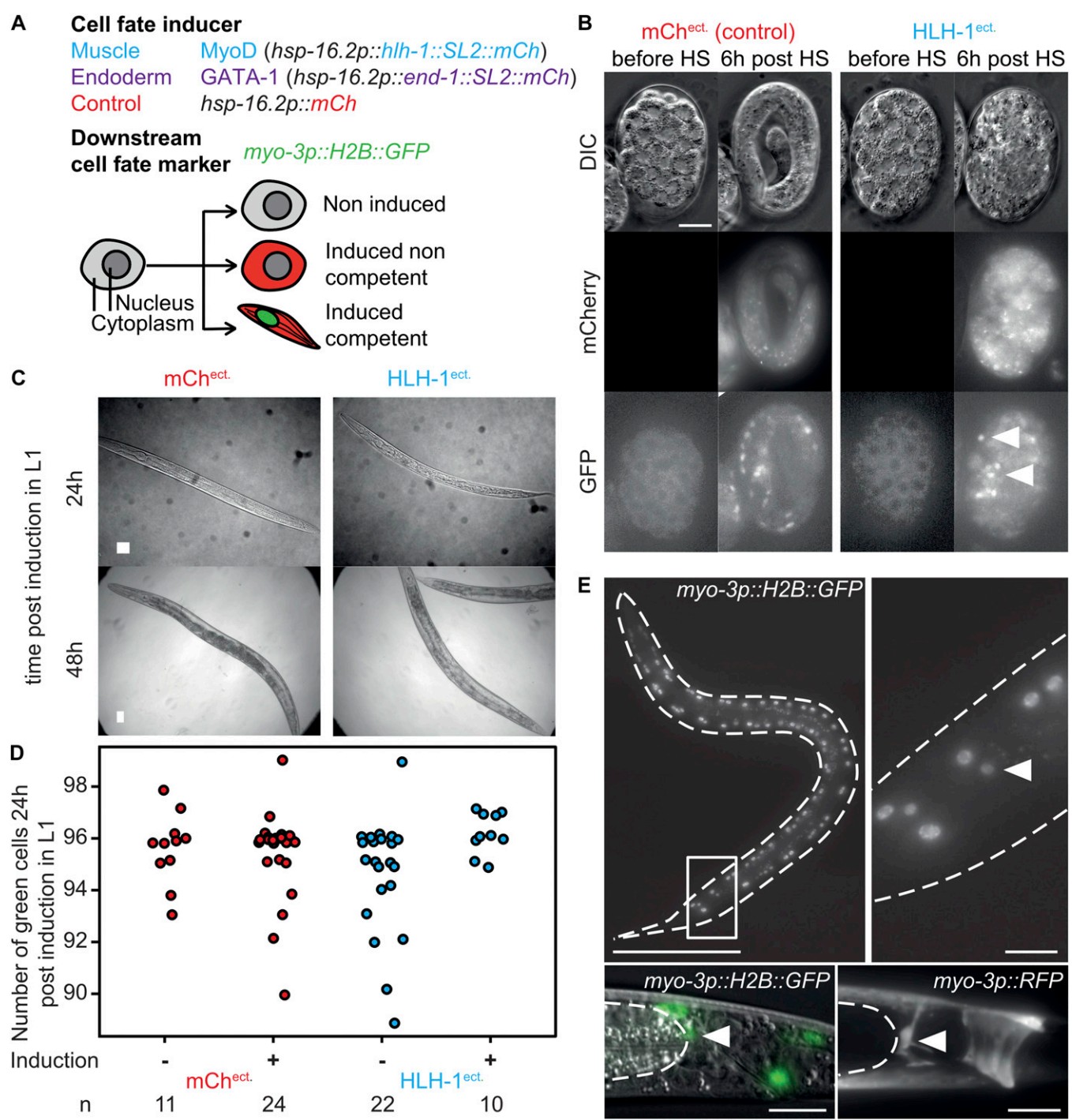

**Figure 1. A single copy system to challenge cell fate highlights robustness of differentiated cells in *C. elegans*.**
**(A)** Single-cell readout cellular plasticity sensor. Cell fate–specifying transcription factors *hlh-1* (MyoD homolog, inducing muscle fate) or *end-1* (GATA-1 homolog, inducing intestinal fate) are induced by HS. Transcription factor ORFs are placed upstream of a trans-spliced *mCherry* ORF, providing a fluorescent readout. A cell fate marker (H2B::GFP) for muscle fate is integrated elsewhere in the genome. All constructs are single-copy insertions. Upon HS, red cytoplasmic fluorescence reports induction whereas green fluorescence reports muscle differentiation. **(B)** Muscle cell fate induction in early embryos (~35 cell stage), DIC, and red and green fluorescence channels before and 6 h post-induction, in a control strain and upon HLH-1 ectopic expression. Upon HLH-1 expression, embryos arrest development and a number of cells express the muscle-specific marker (arrows). Scale 10 μm. **(C)** Brightfield images of worms ectopically expressing either *mCherry* or *hlh-1* 24 and 48 h post-induction. Bar 25 μm. **(D)** Number of GFP::H2B–positive cells of worms in (C). 24 h post-induction. **(E)** Upper left: GFP::H2B signal in an animal ectopically expressing HLH-1, 24 h post-induction (z maximal intensity projection). Bar 100 μm. Upper right: tail region; The additional green fluorescent nucleus is indicated with an arrow. Bar 10 μm. Lower left: DIC/green fluorescence overlay of the tail of an animal of the same strain, imaged 24 h post-induction of HLH-1. Bar 10 μm. Dashed line: gut, white arrow: anal sphincter cell expressing the muscle marker. Lower right: tail region, imaged 24 h post–HLH-1 induction in a strain carrying a cytoplasmic red muscle marker (*myo-3p::RFP*). Bar 10 μm. The cytoplasmic RFP signal outlines the characteristic saddle-like shape of the anal sphincter cell.

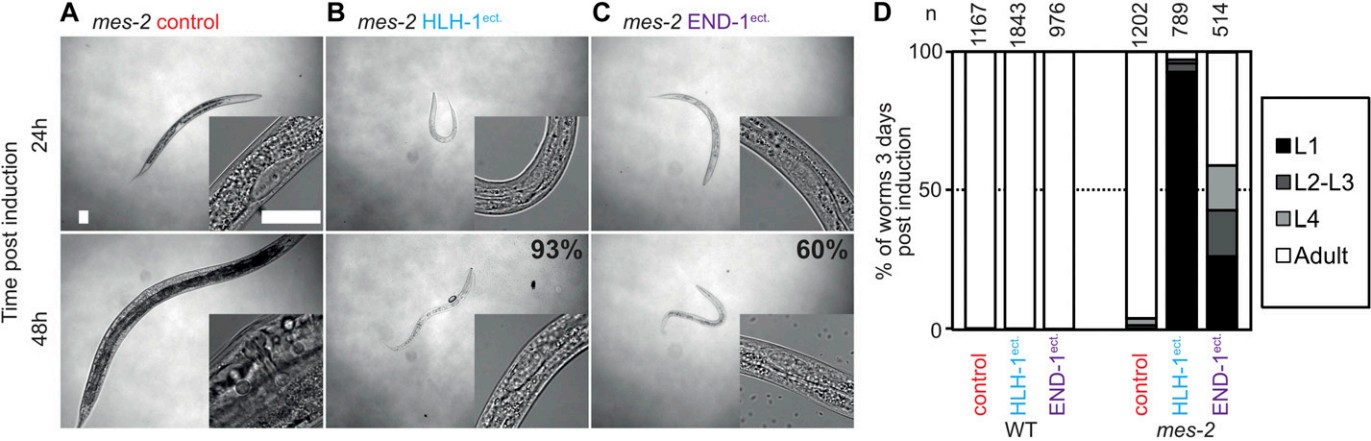

**Figure 2. Absence of the Polycomb protein *mes-2* renders animals sensitive to cell fate challenge.**
**(A)** Brightfield images of *mes-2* animals 24/48 h post-HS. A magnification of the gonad is shown for staging purposes (bars 25 μm). **(B)** Brightfield images of *mes-2* animals ectopically expressing HLH-1 24/48 h post-induction. **(C)** Brightfield images of *mes-2* animals ectopically expressing END-1 24/48 h post-induction. **(D)** Scoring of animal development 3 d post-induction at the first larval stage. Proportions of the populations are shown, key as indicated; L, larval stage.

and alive, the developmentally arrested animals remain small and die without resuming development 7–10 d later. Developmental arrest is observed upon HLH-1[ect.] induction not only at the L1 stage, but also at all stages. As for HLH-1, expression of END-1[ect.] leads to developmental arrest in 60% of *mes-2* F2 L1 animals, although they arrest at later stages of development (Fig 2C and D). Arrest is, however, as stringent as for HLH-1[ect.] because these animals die within 3–7 d without resuming development.

### Additional cells expressing the muscle marker appear upon muscle induction in *mes-2* animals

In wild-type animals, apart from the anal sphincter cell, HLH-1[ect.] expression has no effect on the number of cells expressing the muscle marker. As development proceeds, the number of muscle cells increases from one larval stage to the next (Sulston & Horvitz, 1977) (Fig 3A, WT HLH-1[ect.]). *mes-2* animals upon HLH-1[ect.] expression show 82 to 97 green nuclei per animal, although these are arrested at the L1 stage (Fig 3A *mes-2* HLH-1[ect.]). In contrast, END-1[ect.] does not lead to the appearance of additional cells positive for the muscle marker, although it induces a similar larval arrest (Fig 3A, *mes-2* END-1[ect.]). Conversely, ectopic expression of END-1 in *mes-2* mutants led to a small yet significant increase in endodermal cells (Fig S4A).

To precisely map the location of the cells expressing muscle markers, we genetically reduced the high autofluorescence of gut granules using the *glo-1(zu391)* mutation. Upon HLH-1[ect.] expression, *glo-1 mes-2* behave similarly to *mes-2* animals (Fig S4B and C). We segmented the animals in five sections and counted the number of muscle marker–positive cells in the individual sections (Figs 3B and S4D). Compared with controls, of these five sections, the dorsal side and the ventral gonad to rectum sections showed additional cells (Figs 3B and S4D, arrows). On the clearly delimited ventral gonad to rectum region, instead of the expected 7–8 body wall muscle cells (four on either side left/right, mean 7.3, median 7, n = 15), up to 17 positive nuclei could be

scored (mean 11.9, median 11, n = 29, example in Fig S4D, *mes-2 glo-1* HLH-1[ect.]).

### A number of cell lineages re-enter proliferation upon cell fate challenge in *mes-2* animals

The cells expressing the muscle marker could either come from cell division of muscle cells present in this region or from a trans-differentiation event of other cells types. The precisely defined region between the vulva and the rectum contains a limited number of cell types in L1 larvae: body wall muscles, seam cells of the V lineage, intestinal cells (E lineage), the M progenitor cell, ventrally located P cells (Sulston & Horvitz, 1977). To discriminate between the two hypotheses above, we used red nuclear markers for the V, P, and E lineages. None of these markers showed co-localization with the muscle marker, demonstrating that the additional cells expressing the muscle marker likely originate from division of the muscle cells. This is unexpected, as in normal animals or *mes-2* control animals, these cells are fully post-mitotic (Sulston & Horvitz, 1977). Together, it suggests that cell fate challenge induces cell division in normally quiescent, post-mitotic muscle cells. Interestingly, of the four lineages tested, three of them (P, V, M) proliferated, with additional cells expressing the lineage-specific markers compared with the expected cell numbers in L1 larvae (Fig 3C). These cells did not co-express the muscle marker with their original cell fate marker. Two to five cells expressing the M lineage markers were observed 3 days post–cell fate challenge, for an expected unique M cell. The P cell marker showed a clear increase from the six expected cells to an average of 18 cells between the gonad and the rectum. These cells were moreover grouped by two or four, suggesting successive rounds of cell divisions. Finally, a chromatin-bound nuclear marker for seam cells showed that many seam cells in arrested L1 animals were blocked in anaphase with an average of 12 arrested divisions for 21 expected seam cells (Fig 3D and E). This unscheduled cell division was not specific for HLH-1[ect.] but also observed for arrested animals

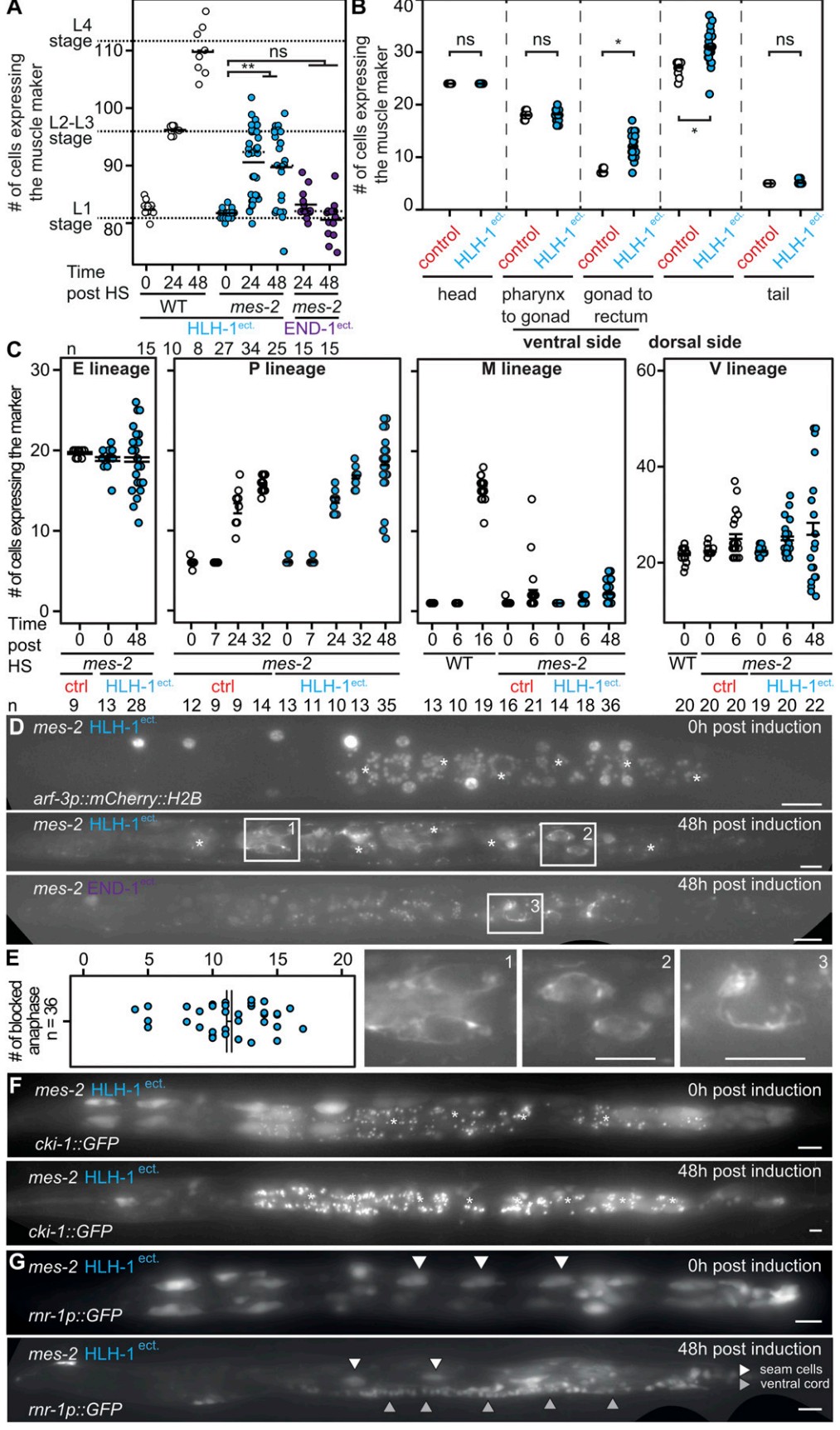

Figure 3. Cell fate challenge in the absence of Polycomb leads to hyperplasia in multiple tissues.
**(A)** Number of GFP::H2B–positive nuclei in wild-type and *mes-2* animals upon ectopic expression of either HLH-1 or END-1 post-induction in the first larval stage. Short solid line: mean; dashed line: median. Wilcoxon test *P*-value 2.57 × 10$^{-09}$. **(B)** Number of GFP::H2B–positive nuclei in different body parts of the worm, in *mes-2 glo-1* control animals and animals ectopically expressing HLH-1. Wilcoxon test *P*-value 1.77 × 10$^{-07}$ (gonad to rectum) and 4.5 × 10$^{-06}$ (dorsal side). **(C)** Number of cells expressing markers for the E, P, M, and V lineages during development and upon HLH-1$^{ect.}$ induction. **(D)** Red fluorescence signal of the V lineage in *mes-2* worms expressing HLH-1$^{ect.}$ or END-1$^{ect.}$ before and 48 h post-induction in *mes-2* worms. Bar 10 μm. **(E)** Number of anaphases bridges in *mes-2* arrested animals upon ectopic expression of HLH-1 48 h post-induction. **(1, 2, 3)** Magnification of anaphases bridges in animals as (C). Bar 10 μm. **(F)** *cki-1::GFP* expression and localization in *mes-2* worms before and 48 h upon HLH-1$^{ect.}$ induction. Bar 10 μm. **(G)** *rnr-1p::GFP* expression and localization in *mes-2* worms before and 48 h after HLH-1$^{ect.}$ induction. White arrows: seam cells, gray arrows: neurons of the ventral cord (P lineage). Bar 10 μm.

after END-1[ect.] induction. This suggests that cells of the V lineage enter mitosis with unreplicated DNA, leading to mitotic catastrophes. Therefore, we conclude that cell fate challenge unexpectedly induces proliferation rather than cell fate change. The unscheduled cell division of V cells is likely to be the cause for the observed larval arrest as the hypoderm is probably no longer functional.

A major regulator of cell cycle is the *C. elegans* homolog of p27/Kip1 CKI-1 (Hong et al, 1998). We therefore sought to quantify expression levels of the gene using a previously published transcriptional reporter (Hong et al, 1998). Correlated with the observed hyperplasia in multiple cell types, levels of cyclin kinase inhibitor CKI-1 decreased upon HLH-1[ect.] induction (Fig 3F). Conversely, a GFP transgene driven by the *rnr-1* ribonucleotide reductase promoter normally expressed in replicating cells showed widespread expression only upon HLH-1[ect.] induction (Fig 3G), in particular in the P lineage. In conclusion, although dispensable for normal fate specification, PRC2 and H3K27me are essential to stabilize cell fate upon perturbations and protect cells from unscheduled cell division.

## Knock-down of Notch signaling pathway components rescues larval arrest and aberrant cellular proliferation

To further understand the function of Polycomb upon cell fate challenge, we reasoned that knock-down of factors potentially enhancing plasticity could suppress proliferation defects and larval arrest induced by HLH-1[ect.] or END-1[ect.] in *mes-2* animals. We therefore screened a library of previously characterized genes involved in cellular plasticity in nematodes using RNAi (Tables S1). Most RNAi had no effect and a large majority of *mes-2* animals arrested development upon HLH-1[ect.] expression (Fig S5). In contrast, *lin-12(RNAi)* rescued 55% of the population, which developed to adulthood (Fig 4A and B). LIN-12 is one of the two Notch receptor homologs. Knock-down of the second homolog *glp-1* had no effect, in agreement with its role in the germline and during embryogenesis (Priess, 2005). Similar to *lin-12* RNAi, knock-down of the Notch ligand components (*lag-2, dos-2,* and *dos-3*), the Notch co-ligands (*osm-7* and *osm-11*) and some Notch target genes (*lip-1, lst-1,* and *lst-4*) rescued larval arrest, although to a lesser degree than *lin-12(RNAi)* (Fig 4B). Rescue from larval arrest by *lin-12(RNAi)* was also observed upon END-1[ect.] induction, in which we could detect a reduction from 18 to 10% of arrested L1 worms by *lin-12* RNAi (n = 582, 664). Moreover, knock-down of *lin-12,* in addition to the larval arrest rescue, reverted the cellular proliferation phenotypes of arrested worms described above: animals showed no significantly different number of muscle cells when compared with control *mes-2* animals (Fig 4C). Similarly, no premature cell division was observed in the seam cell lineage (Fig S6).

## The Notch ligand LAG-2 is up-regulated upon cell fate challenge

Our data demonstrate that the Notch pathway sensitizes animals to cell fate challenge, leading to cell proliferation, mitotic catastrophes, and larval arrest. We therefore assayed whether LIN-12[Notch] or the Notch ligand LAG-2 would be up-regulated in *mes-2* animals before and after HLH-1[ect.] induction. We used GFP fusions for the Notch receptor LIN-12 and a transcriptional fusion for LAG-2 (Singh

et al, 2011; Sarov et al, 2012). For LIN-12[Notch], no significant up-regulation of the abundance of the receptor could be observed in *mes-2* control or HLH-1[ect.] animals, although it has to be noted that the expression levels of this protein are quite low outside of the developing vulva, making precise quantifications difficult (Fig S7A and B). No fluorescence could be observed in the M lineage progenitor, whereas previous lacZ stainings detected the expression of a transgene driven by the *lin-12* promoter (Wilkinson & Greenwald, 1995). In contrast, a GFP transgene driven by the LAG-2 promoter showed at least a global two-fold up-regulation upon HLH-1[ect.] induction (Figs 4D and S7C). Although this transgene is normally mostly expressed in neurons, HLH-1[ect.] induction led to high fluorescence levels in the gut and the posterior part of the animal (Figs 4D and S7C). Together, we conclude that the Notch receptor is not over-expressed upon HLH-1[ect.] induction, but Notch signaling is activated by increased levels of the Notch ligand LAG-2. Indeed, analysis of chromatin immunoprecipitation data upon HLH-1[ect.] induction in embryos shows binding of this transcription factor to the promoter of the Notch ligand *lag-2,* and the promoters of the nuclear factors *sel-8* and *lag-1,* both involved in LIN-12[Notch]-mediated gene activation (Fukushige & Krause, 2005). Our results demonstrate an up-regulation of the Notch ligand upon cell fate challenge in *mes-2* animals, which ultimately leads to cell proliferation trough Notch pathway activation. To discriminate whether *mes-2* protects *lag-2* itself from activation or rather affects downstream Notch target genes, we tested whether *lag-2* activation upon cell fate challenge in wild-type animals using the same *lag-2::GFP* transcriptional reporter. We observe that GFP levels are initially similarly up-regulated by HLH-1[ect.] in wild-type animals and in *mes-2* ones (Fig 4D and E). Decrease of the signal in wild-type animals at later time points is because of the fact that these animals are growing, in contrast to *mes-2* developmentally arrested ones. The up-regulation of LAG-2 is therefore a consequence of cell fate challenge, whereas *mes-2* likely protects Notch target genes from unscheduled activation (Fig S7D and E).

## Notch pathway activation in wild-type animals leads to a small proportion of larval arrests

Our results show that LIN-12[Notch] signaling is increased upon cell fate challenge, whereas larval arrest is rescued by knock-down of the Notch components, suggesting that this aberrant activation of this pathway is causal for the cellular phenotypes and the larval arrest of *mes-2* animals. We therefore wanted to test whether continuously increasing Notch signaling in wild-type animals could lead to larval arrest. To this end, we genetically increased LIN-12[Notch] activity using a gain-of-function *lin-12* allele (Fig 4F). Notch pathway activation led to larval arrest in 2% of the animals, a phenotype which was never observed in animals with normal Notch signaling after HLH-1[ect.] induction ($P = 5 \times 10^{-15}$). This phenotype is independent of HLH-1[ect.] expression as a similar proportion of *lin-12(gf)* animals experienced larval arrest upon cell fate challenge. This strongly supports both the fact that LIN-12[Notch] signaling is downstream of cell fate challenge, and that the function of the Polycomb complex is to protect Notch target genes from unscheduled activation.

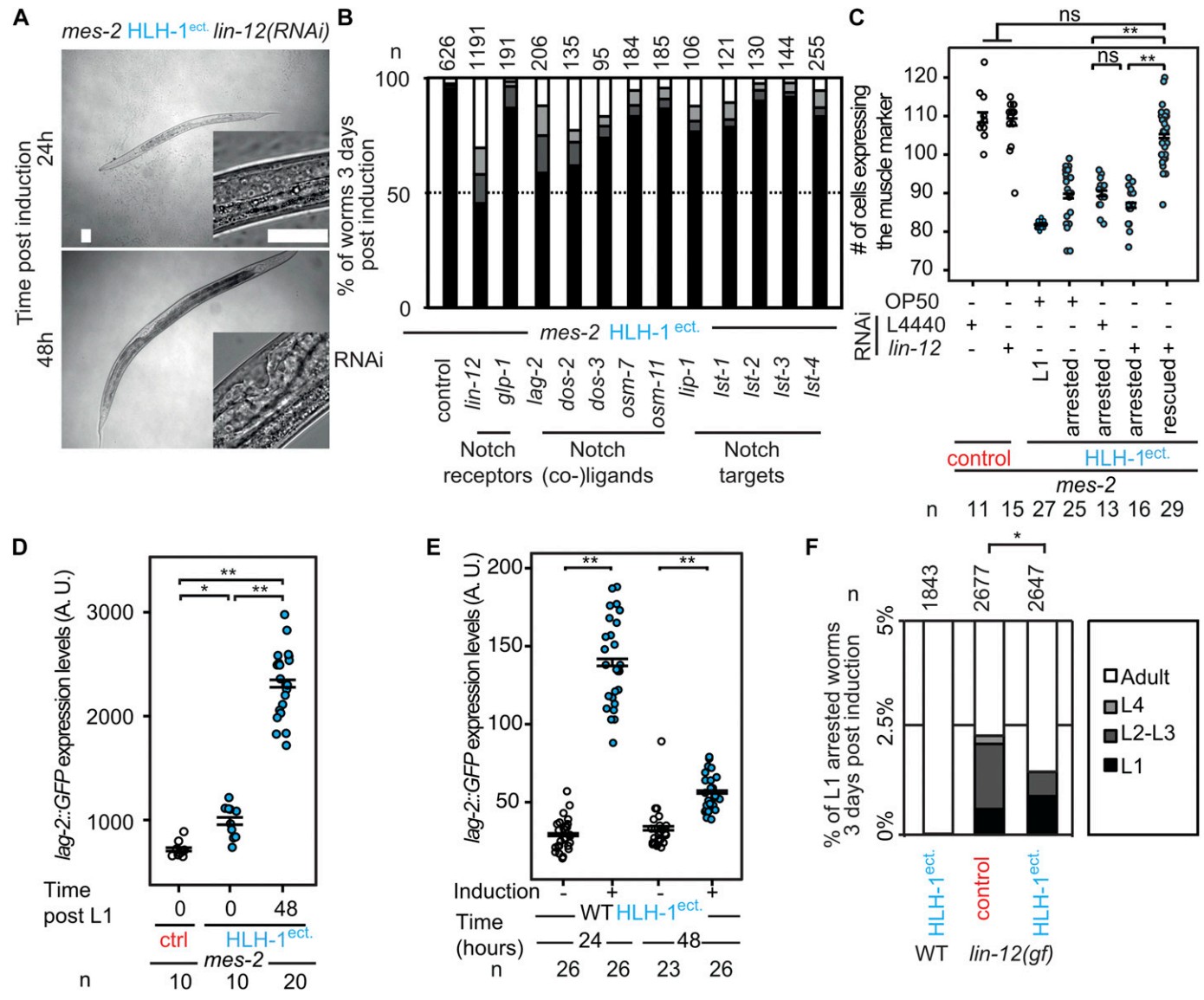

**Figure 4. Hyperplasia is due to Notch pathway activation upon cell fate challenge.**
**(A)** Brightfield images of rescued *mes-2* mutant animals grown on *lin-12(RNAi)* ectopically expressing HLH-1 24/48 h post-induction. Next to the picture of the entire animal (bar 25 *μ*m), a magnification of the gonad/vulva is shown for staging purposes (bar 25 *μ*m). **(B)** Scoring of development 3 d post-induction of HLH-1 expression at the first larval stage in *mes-2* animals fed on RNAi for the indicated genes. Color key in (F). **(C)** GFP::H2B–positive nuclei in control animals (white) and a strain ectopically expressing HLH-1 (blue) in a *mes-2* genetic background fed on *L4440(RNAi)* and *lin-12(RNAi)*. Comparison with arrested L1 animals 3 d post–HLH-1 induction. Wilcoxon test *P*-value $1.86 \times 10^{-06}$ and $1.05 \times 10^{-04}$ between *lin-12(RNAi)* rescued animals and control *L4440(RNAi)*, and *lin-12(RNAi)*–arrested animals and control *L4440(RNAi)*, respectively. **(D)** Quantification of the fluorescent signal from the *lag-2p::GFP* transgene in control (white) and upon HLH-1[ect.] induction (blue) in a *mes-2* background before and after induction. Wilcoxon test *P*-values * = 0.0001, ** = $6.6 \times 10^{-8}$. **(E)** Quantification of the fluorescent signal from the *lag-2p::GFP* transgene in wild-type animals. Comparison between animal control (white) and upon HLH-1[ect.] induction (blue). Wilcoxon test *P*-values ** = $6.05 \times 10^{-10}$, ** = $2.35 \times 10^{-7}$. The differences in *lag-2p::GFP* expression levels in (D) and (E) are due to image acquisition with different magnification because of animal size differences (F). Scoring of arrested animals 3 d post-induction in control strain and upon HLH-1[ect.] expression at the first larval stage in wild-type animals or wild-type animals with the gain-of-function *lin-12(n950)* mutation. Proportions of the populations are shown.

## Starvation status determines sensitivity to cell fate challenge

If LIN-12[Notch] signaling transduces a signal which leads to hyperplasia, situations where LIN-12[Notch] signaling is blocked should rescue transdifferentiation-induced larval arrest. One such situation has been described in vulval precursor cells (P lineage) of the dauer larvae, a starvation resistant developmental stage induced by harsh conditions and particularly insensitive to a number of environmental stresses (Golden and Riddle 1982, 1984). Because in dauers, LIN-12[Notch] signaling is actively blocked (Karp & Greenwald, 2013), we assayed whether *mes-2* dauers were sensitive to HLH-1[ect.]. After induction, animals transferred to food-containing plates resumed growth normally until adulthood (n = 33). This suggests that the dauer stage is resistant to cell fate

challenge, possibly through the inhibition of LIN-12$^{Notch}$ signaling. During the nematode life cycle, a second starvation and stress-resistant stage called the L1 diapause occurs when larvae of the first larval stage hatch in the absence of food (Baugh, 2013). We therefore assayed whether these animals would be resistant to cell fate challenge as dauers. Upon HLH-1$^{ect.}$ expression, only 3% of the starved L1 *mes-2* animals arrested development compared with 93% in the presence of food similarly to *mes-2* expression, in which almost all worms developed to adulthood (Fig 5A–D). Importantly, the difference between fed and starved animals is not

due to significant differences in the expression levels of the ectopically expressed transcription factors, as fluorescence levels from the trans-spliced *mCherry* 6 h post-HS were similar (Fig 5E). Starved animals in which HLH-1$^{ect.}$ was induced have similar numbers of muscle and seam cells as control animals 48 h post-induction (Fig S8). Rescue from larval arrest by starvation-induced L1 diapause is not merely the result of an environmental stress, as treating fed *mes-2* animals with high salt concentrations or oxidizing H$_2$O$_2$ before HLH-1$^{ect.}$ induction could not rescue the larval arrest phenotype (Fig S9).

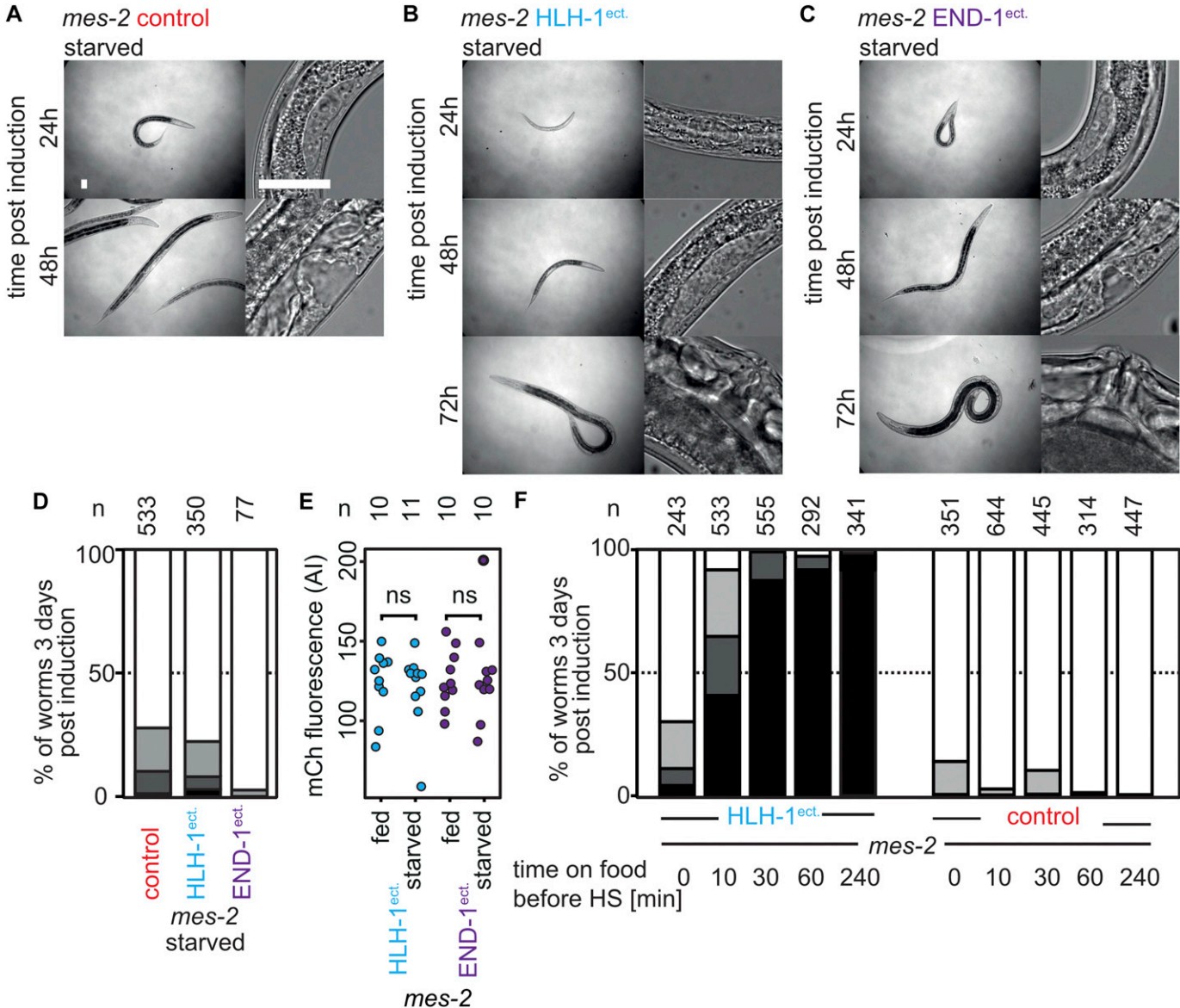

**Figure 5.   Starvation protects Polycomb-defective animals from cell fate challenge.**
**(A)** Brightfield images of *mes-2* mutant animals returned to feeding plates after HS in starved conditions, 24/48 h post-treatment. Magnification as in 2A (bar 25 μm). **(B)** Brightfield images of *mes-2* mutant animals returned to feeding plates after HLH-1$^{ect.}$ expression in starved conditions, 24/48/72 h post-treatment. **(C)** Brightfield images of *mes-2* mutant animals returned to feeding plates after END-1$^{ect.}$ expression in starved conditions, 24/48/72 h post-treatment. **(D)** Scoring of development 3 d post-induction at the first larval stage in starved conditions. Color key as in Fig 2D. **(E)** Quantification of the mCherry fluorescence from the trans-spliced ORF 6 h post-induction in fed or starved animals. **(F)** Scoring of animal development 3 d post-induction at the first larval stage after feeding for given times. Color key as in Fig 2D.

This protective effect of starvation disappears rapidly and irreversibly when L1 animals are fed. Placing starved animals on food for 30 min before HLH-1[ect.] induction is sufficient to elicit almost complete arrest of the population at the L1 stage (Fig 5F, HLH-1[ect.]). Moreover, *mes-2* animals in which ectopic expression of HLH-1 was induced in the starved L1 stage (no developmental arrest) and left to grow 24 h on food to reach L2/L3 stage before induction of HLH-1 arrested at the L3/L4 stage (93% of n = 111). Similarly, short-term starvation (6 h at the L2/L3 stage) was unable to evoke this protective effect (92% arrested animals, n = 120). Altogether, our results point to a protective role for diapause upon cell fate challenge.

# Discussion

Cell fate maintenance is a key feature for life of multicellular organisms, with important health implications when impaired. The nematode *C. elegans* has proven to be a very useful organism to uncover barriers to reprogramming using either cell fate challenge or natural transdifferentiation (Hajduskova et al, 2012). Here, using a novel, tractable single-copy system for cell fate challenge insensitive to multicopy array heterochromatinization, we demonstrate with single-cell resolution that fully differentiated animals are highly resistant to transdifferentiation: only a single cell transiently expresses markers for the induced cell fate. Moreover, we show that ablation of the two classical heterochromatin marks H3K9 and H3K27 methylation have different effects: whereas animals lacking H3K9 methylation are identical to wild-type, animals without H3K27 methylation are exquisitely sensitive to cell fate challenge and terminally arrest development. Unexpectedly, this larval arrest is not due to transdifferentiation of a large number of cells, but rather due to unscheduled cell divisions of many tissues of different embryonic origin or hyperplasia. These include cells in which the genome was not previously replicated, leading to mitotic catastrophe. Although the mechanism by which H3K27 methylation protects animals from unscheduled cell division is unclear, one could speculate that the histone modification precludes access to the transcription factor target genes driving cell cycle progression. Indeed, HLH-1 has been shown previously to act together with chromatin remodelers to counteract Polycomb-mediated transcriptional repression (Ruijtenberg & van den Heuvel, 2015). In the absence of H3K27 methylation deposited by Polycomb, expression of the normally suppressed cell cycle progression genes would lead to the observed hyperplasia. Our data provide the first evidence of a widely distributed somatic cell hyperplasia, suggesting that the Polycomb group (PcG) is part of the machinery which maintains cells in a quiescent state. Interestingly, epigenetic regulators are the most commonly mutated genes in human cancers (Piunti & Shilatifard, 2016). Moreover, In flies, similar hyperplasia accompanied by tumour formation has been described to appear spontaneously upon deletion of PcG genes in actively growing tissues (Classen et al, 2009; Martinez et al, 2009; Beira et al, 2018). This spontaneous occurrence correlates with the fact that PcG genes are essential for fly development. In contrast, nematodes achieve full development in the absence of maternal and zygotic loads of *mes-2*, and hyperplasia is only observed upon induction of cell

fate challenge, suggesting that other systems keep cell proliferation in check in the absence of Polycomb.

Using a candidate RNAi screen to uncover the regulators of induced transdifferentiation sensitivity, we found that knocking down components of the Notch intercellular signaling pathway suppresses the sensitivity of Polycomb mutants to cell fate challenge. Suppression occurs at the level of the organism because developmental arrest is abolished, and at the cellular level as no unscheduled mitosis is observed, further supporting the notion that abortive mitosis would be the cause for developmental arrest. We show that activation of the Notch pathway is a consequence of the induced overexpression of LAG-2, the ligand of the LIN-12[Notch] receptor (Greenwald, 2005). Interestingly, Notch signaling is essential to endow the endodermal Y cell with the competence for natural transdifferentiation (Jarriault et al, 2008) and favors direct germline cell reprogramming, again by antagonizing Polycomb-mediated silencing (Seelk et al, 2016). This interplay between cell signaling and PcG chromatin modifications has been repeatedly described in *Drosophila*. Deletion of PcG gene *polyhomeotic* (*ph*) leads to JNK, JAK/STAT, and Notch pathway activation (Classen et al, 2009; Martinez et al, 2009; Beira et al, 2018). In particular, members of the Notch pathway, but not the Notch ligand Delta, are derepressed in *ph* mutants (Martinez et al, 2009). Our results therefore suggest a high level of conservation of the interplay between Notch and Polycomb between flies and worms to regulate cell proliferation.

LIN-12[Notch] signaling was previously shown to accelerate dauer and L4 quiescence exit, during which animals undergo major organismal reorganization, cell fate changes, and/or cell divisions (Roy et al, 2002; Singh et al, 2011). This is comparable with our L1 diapause exit experiments, as in all three situations, cells prepare for subsequent cell fate modifications and cellular division. The exit of the L1 diapause induced by animal feeding acts as a fast switch, inducing sensitivity to ectopic expression of cell-fate–inducing transcription factors. A similar cell division phenotype has been previously observed in mutants of the transcription factor *daf-16* which transduces insulin signaling in the nucleus. In *daf-16* mutants, extended L1 diapause leads to a decrease in CKI-1 expression and premature seam cell divisions in arrested animals (Baugh & Sternberg, 2006). This suggests that the insulin signaling pathway transduces the sensation of food in the environment to change cell plasticity, rendering the animals sensitive to cell fate challenge in the sensitized *mes-2* background. Testing the phenotypes of these mutants, in particular *daf-2* and *daf-16*, and a combination of those would prove this hypothesis.

Our results show that Notch signaling could be downstream of the food-sensing pathway, mediating cell plasticity at the cellular level. It remains to be elucidated whether cell cycle progression inhibition in Notch knock-down animals, which rescues larval arrest induced by cell-fate challenge, is cell autonomous or not. In other developmental systems, the highly conserved Notch signaling pathway has been involved in a variety of cell fate decisions, including cell division and cell differentiation (Totaro et al, 2018). Moreover, Notch mutations are found in many human tumors (Mutvei et al, 2015). Our results highlight starvation as an unexpected regulator of Notch signaling, providing an interesting and actionable path for Notch regulation.

# Materials and Methods

### General worm methods

Unless otherwise stated, *C. elegans* strains were grown on NG2 medium inoculated with OP50 bacterial strain at 22.5°C. The dominant *lin-12(n950)* gain of function mutation was introduced as in (Katic et al, 2015).

### Synchronization and TF induction in embryos and larvae

For embryos, wild-type or F1 *mes-2* gravid adults were dissected in M9, one and two cell stage embryos were transferred to a 2% agar pad and incubated at 24°C until they reached the desired stage (as described in Kiefer et al [2007]). The developmental stage was verified before HS by imaging and the expression of the transcription factor was induced by 10 min HS at 33°C in a PCR thermocycler.

For larvae synchronization, wild-type worms were synchronized either by letting gravid adults lay eggs for 2–4 h on plates or by bleaching gravid adults. Embryos were then left to hatch overnight at 22.5°C. *mes-2* animals were synchronized as described in the Materials and Methods section of the Supplementary Information. Animals were heat-shocked in a 33°C water bath for 30 min, before transferring them to a fresh plate seeded with OP50 and incubated at 22.5°C.

### Evaluation of the development stage of the animals

The developmental stage of wild-type and *mes-2* worms was evaluated, respectively, 2 or 3 d post-induction. Supposedly arrested animals were moved to new plates to assess the reality of the developmental arrest. Worms were scored according to their size and the appearance of the vulva and gonad into L1, L2-3, L4, and adult. The *mes-2* worms were checked again at day 7 post-induction to verify the results.

### Imaging and image analysis

The worms were imaged on an iMIC (FEI Munich GmbH) equipped with filters for DIC, brightfield *mCherry*, and *GFP* detection and an ORCA-R2 CCD camera (Hamamatsu). Whole worms were imaged with 20–60× magnification in z stacks with a 0.5–1 $\mu$m distance between planes. Fiji was used for picture stitching and GFP-positive nuclei counting. *mCherry* expression was measured selecting regions of interests around animals and in the background. Final statistical comparisons were made in Microsoft Excel and R. Experiments performed at least twice with a minimum of three samples.

### RNAi experiments by feeding

Double-stranded expressing bacteria from the (Ahringer library, Kamath & Ahringer, 2003) seeded on NG2 plates were used for RNAi experiments. Heterozygous L3-L4 *mes-2* animals were moved onto RNAi plates and an *mes-2* homozygous F2 generation was used for the experiments. After induction of transcription factor expression, worms were moved to RNAi plates.

# Supplementary Information

# Acknowledgements

The authors wish to thank members of the Meister laboratory for numerous discussions; Julie Campos and Dr. Jennifer Semple for expert technical help. We wish to thank Dr. Rafal Ciosk for the muscle marker, Dr. Michalis Barkoulas for the chromatin-bound seam cell marker and Dr. Iskra Katic for help with CRISPR-mediated mutagenesis. Some strains were provided by the CGC, which is funded by NIH Office of Research Infrastructure Programs (P40 OD010440). The Meister laboratory is supported by the Swiss National Science Foundation (SNF assistant professor grant PP00P3_133744/159320, 31003A_176226), the Swiss Foundation for Muscle Diseases Research and the University of Bern.

### Author Contributions

F Coraggio: conceptualization, formal analysis, investigation, and writing—original draft, review, and editing.
R Püschel: conceptualization, formal analysis, investigation, and writing—original draft.
A Marti: investigation.
P Meister: conceptualization, data curation, formal analysis, funding acquisition, investigation, project administration, and writing—original draft, review, and editing.

### Conflict of Interest Statement

The authors declare that they have no conflict of interest.

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
