## [Reviewer comments · Life Science Alliance]

Life Science Alliance

Polycomb and Notch signaling regulate cell proliferation potential during *C. elegans* life cycle

Francesca Coraggio, Ringo Püschel, Alisha Marti, and Peter Meister
DOI: 10.26508/lsa.201800170

Corresponding author(s): Peter Meister, Institute of Cell Biology

Review Timeline:

Submission Date:	2018-08-23
Editorial Decision:	2018-10-08
Revision Received:	2018-12-14
Editorial Decision:	2018-12-17
Revision Received:	2018-12-17
Accepted:	2018-12-18

Scientific Editor: Andrea Leibfried

Transaction Report:

October 8, 2018

Re: Life Science Alliance manuscript #LSA-2018-00170-T

Dr. Peter Meister
Institute of Cell Biology
University of Bern Baltzerstrasse 4
Bern CH-3012
Switzerland

Dear Dr. Meister,

Thank you for submitting your manuscript entitled "Polycomb and Notch signaling regulate cell proliferation potential during *C. elegans* life cycle" to Life Science Alliance. Please excuse the delay in getting back to you. We were waiting for a report on your work, and this delayed the process. The manuscript was assessed by expert reviewers, whose comments are appended to this letter.

As you will see, both reviewers appreciate your analysis and support publication of a revised version of your manuscript in Life Science Alliance. The revision expected seems minor and straightforward to perform. We would thus like to invite you to submit a revised version, addressing the points raised by the two reviewers by changing the manuscript text and by adding the two experiments requested by reviewer #1.

Thank you for this interesting contribution to Life Science Alliance. We are looking forward to receiving your revised manuscript.

Sincerely,

- A letter addressing the reviewers' comments point by point.
- An editable version of the final text (.DOC or .DOCX) is needed for copyediting (no PDFs).
- High-resolution figure, supplementary figure and video files uploaded as individual files: See our detailed guidelines for preparing your production-ready images, <http://life-science-alliance.org/authorguide>
- Summary blurb (enter in submission system): A short text summarizing in a single sentence the study (max. 200 characters including spaces). This text is used in conjunction with the titles of papers, hence should be informative and complementary to the title and running title. It should describe the context and significance of the findings for a general readership; it should be written in the present tense and refer to the work in the third person. Author names should not be mentioned.

B. MANUSCRIPT ORGANIZATION AND FORMATTING:

Full guidelines are available on our Instructions for Authors page, <http://life-science-alliance.org/authorguide>

Reviewer #1 (Comments to the Authors (Required)):

In this very nice paper, Coraggio et al. describe how ectopic expression of a selector gene does something quite unexpected. Expectedly, misexpression in a wild-type background doesn't do much and, also expectedly, genetic manipulation of chromatin state, does now permit the selector gene

to do "stuff" - but what it does is unexpected. Rather than reprogramming the identity of other cells (as observed in other reprogramming paradigms) the selector gene apparently promotes cell cycle re-entry and proliferation. It's very laudable that the authors bother to test whether ectopic gfp-positive cells are "reprogrammed cell" vs. new, overproliferated cells (there's unfortunately a lack of stringency about this in the literature). This overproliferation phenotype is in turn caused by apparent activation of the Notch pathway. This is very cool. As a side-note, kudos for nice quantification of phenotypes.

Anyway, this is a really interesting paper and I enthusiastically recommend publication once a number of loose ends are tied up:

Major:

1) the authors provide some strong suggestive evidence that cells indeed reenter the cell cycle, i.e. that there is overproliferation. While the evidence is highly suggestive, the authors haven't put the last nail into the coffin, i.e. they never really explicitly show cell division. This really shouldn't be too hard - just spend a few hours on the microscope and observe cell divisions by Nomarski.

Alternatively, the van der Heuvel lab has put a nice marker, *mcm-4::gfp* (or *rfp*, I forgot) out onto the market that allows to directly look at proliferation. I think the authors should be trying one of those two things.

2) the genetics in the manuscript is strong, with the one exception of the *lin-12* analysis, which I find to over-rely in RNAi which always tends to make me a little nervous. There's no reason why the authors can't test a good *lin-12* mutant here.

Minor:

1) line 108 "epigenetically close to a muscle" ? I presume the authors means "lineally", as "epigenetically" makes absolutely not sense in this context.

2) line 180 ("Interestingly...") should be reworded. It's a little unclear, I guess what the authors are saying is that *hlh-1*ectopic minus PRC induces proliferation in P,M WITHOUT marker expression, but that isn't quite clearly phrased.

Reviewer #3 (Comments to the Authors (Required)):

In their study, Coraggio and colleagues reveal a very interesting relationship of the Notch signaling pathway and the chromatin-regulating complex PRC2 in *C. elegans*. By challenging cells in vivo with overexpression of the muscle fate-inducing transcription factor HLH-1 in PRC2 (*mes-2*) mutants they found that animals undergo a developmental arrest. Interestingly, the authors found by performing an RNAi screen that the Notch signaling pathway is required for this arrest. The reason for the observed arrest appears to be unscheduled cell divisions suggesting that PRC2 has a role in preventing such un-wanted extra cell divisions. Moreover, the authors demonstrate that the Notch signaling pathway might be inhibited upon starvation of developing worms as the effects of the cell fate challenges are suppressed. Their results suggest that starvation has an impact on Notch signaling to stabilize cell fates.

Overall, this is a very good study revealing important aspects of cell fate regulation and implication of different signaling pathways in combination with the metabolic state of an organism. The experiments have been performed accurately and the conclusions are plausible. I support publication of this manuscript.

Minor comments:

The authors should consider using the term 'transdifferentiation' more carefully. It is being used several times in the abstract. However, it seems that there are in fact no real, or at least stable, transdifferentiation events. The gain of additional muscle marker-positive cells is due to proliferation and the anal sphincter cell shows only transiently the muscle gene expression. I'd suggest at least using the term as 'transient transdifferentiation' or similar.

With regard to the effects of the endoderm inducer END-1, it would be good to know if the authors tested whether there are also additional cells showing an intestinal fate reporters (e.g. *elt-2::gfp*).

^b
**UNIVERSITÄT
BERN**

Phil. Nat. Fakultät

**Institute of Cell Biology
SNF Prof. Dr. Peter Meister**

Tel. + 41 31 631 46 09
E-Mail: peter.meister@izb.unibe.ch

Response to reviewers

Reviewer #1 (Comments to the Authors (Required)):

In this very nice paper, Coraggio et al. describe how ectopic expression of a selector gene does something quite unexpected. Expectedly, misexpression in a wild-type background doesn't do much and, also expectedly, genetic manipulation of chromatin state, does now permit the selector gene to do "stuff" - but what it does is unexpected. Rather than reprogramming the identity of other cells (as observed in other reprogramming paradigms) the selector gene apparently promotes cell cycle re-entry and proliferation. It's very laudable that the authors bother to test whether ectopic gfp-positive cells are "reprogrammed cell" vs. new, overproliferated cells (there's unfortunately a lack of stringency about this in the literature). This overproliferation phenotype is in turn caused by apparent activation of the Notch pathway. This is very cool. As a side-note, kudos for nice quantification of phenotypes.

We would like to thank reviewer #1 for his positive and encouraging comments.

Anyway, this is a really interesting paper and I enthusiastically recommend publication once a number of loose ends are tied up:

Major:

*1) the authors provide some strong suggestive evidence that cells indeed reenter the cell cycle, i.e. that there is overproliferation. While the evidence is highly suggestive, the authors haven't put the last nail into the coffin, i.e. they never really explicitly show cell division. This really shouldn't be too hard - just spend a few hours on the microscope and observe cell divisions by Nomarski. Alternatively, the van der Heuvel lab has put a nice marker, *mcm-4::gfp* (or *rfp*, I forgot) out onto the market that allows to directly look at proliferation. I think the authors should be trying one of those two things.*

We have been testing both types of experiments. Long term immobilization (3 days) of the animals on levamisole (to allow microscopic scoring) or microfluidic devices impaired any cell division. We therefore turned to molecular markers of DNA replication. While the *mcm-4::mCherry* strain from the van der Heuvel laboratory gave too weak signals, we were fortunate enough to use in parallel another published array in which GFP is under the transcriptional regulation of the *rnr-1* promoter (*mals103[rnr-1p::GFP unc-36(+)]*). *rnr* stands for ribonucleotide reductase, the essential enzyme which converts ribonucleotides to deoxyribonucleotides, expressed only during the S phase of the cell cycle. In normal animals, *rnr-1* expression is restricted to replicating cells in both the soma or the germline. Accordingly, in both wild-type or *mes-2* mutant L1 larvae, we observe clear

Institute of Cell Biology
SNF Prof. Dr. Peter Meister

Baltzerstrasse 4
CH-3012 Bern

+41 31 631 46 09

peter.meister@izb.unibe.ch
www.izb.unibe.ch

expression in the seam cells. Upon ectopic induction of HLH-1, we observe an activation of the *mnr-1* promoter in many different cell types, including the neurons of the ventral cord. These data are now integrated in figure 3G.

2) *the genetics in the manuscript is strong, with the one exception of the lin-12 analysis, which I find to over-rely in RNAi which always tends to make me a little nervous. There's no reason why the authors can't test a good lin-12 mutant here.*

We created a strain carrying a *lin-12(n941)* null mutation in a *mes-2* null background. As *lin-12(n941)* homozygote animals are sterile, as well as *mes-2(bn11)*, the mutations are rescued by two different genetic balancers. In this context, we assayed the effect of the ectopic expression of HLH-1, comparing it with a control *lin-12(n941)* strain. Unfortunately, the *lin-12(n941) mes-2(bn11)* F2 animals are extremely sensitive to the heat-shock treatment used to induce the expression of HLH-1, with more than 60% of the animals developmentally arrested 3 days post heat shock. In these conditions, if the *lin-12(n941)* mutation would reproduce the *lin-12(RNAi)* phenotype, one would expect at most 20% of non arrested animals after heat-shock. Indeed, we observe that approximately 20% of the animals develop further than the L1 stage (figure for reviewers). We think however that these results are difficult to interpret given the low fitness of the *lin-12 mes-2* mutants. We have therefore decided not to integrate them into the manuscript. If we agree that the RNAi phenotypes should be taken with caution, our conclusion is backed-up by the fact that rescue is observed after knock-done of multiple members of the LIN-12^{Notch} pathway.

We did an additional experiment to further strengthen the data and uncover whether *lag-2* or Notch target genes are protected from activation by *mes-2* mediated H3K27 methylation. To this aim, we performed the cell fate challenge with HLH-1^{ect.} in a strain carrying the *lag-2p::GFP* transgene in an otherwise wild-type background. Strikingly, we observe high levels of fluorescence, suggesting that cell fate challenge similarly activates *lag-2* transcription in both the *mes-2* and the wild-type situation. Therefore, larval arrest and cell proliferation in the *mes-2* background are not a due to *lag-2* activation (which is similar in wild-type and *mes-2* backgrounds), but can be interpreted as the activation of the Notch target genes in *mes-2* animals. We have now integrated these additional results in figure 4/S7 and discussed these.

Minor:

1) *line 108 "epigenetically close to a muscle" ? I presume the authors means "lineally", as "epigenetically" makes absolutely not sense in this context.*

As suggested, we modified epigenetically to lineally.

2) *line 180 ("Interestingly...") should be reworded. It's a little unclear, I guess what the authors are saying is that hlh-1ectopic minus PRC induces proliferation in P,M WITHOUT marker expression, but that isn't quite clearly phrased.*

We re-phrased this sentence to make it clearer. The new sentence reads:

«Interestingly, out of the 4 lineages we tested 3 of them (P, V, M) proliferated, with additional cells expressing the lineage-specific markers compared to the expected cell numbers in L1 larvae (Figure 3C). These cells did not co-express the muscle marker with their original cell fate marker. Therefore, we conclude that cell fate challenge unexpectedly induced proliferation rather than cell fate change.»

Reviewer #3 (Comments to the Authors (Required)):

In their study, Coraggio and colleagues reveal a very interesting relationship of the Notch signaling pathway and the chromatin-regulating complex PRC2 in C. elegans. By challenging cells in vivo with overexpression of the muscle fate-inducing transcription factor HLH-1 in PRC2 (mes-2) mutants they found that animals undergo a developmental arrest. Interestingly, the authors found by performing an RNAi screen that the Notch signaling pathway is required for this arrest. The reason for the observed arrest appears to be unscheduled cell divisions suggesting that PRC2 has a role in preventing such un-wanted extra cell divisions. Moreover, the authors demonstrate that the Notch signaling pathway might be inhibited upon starvation of developing worms as the effects of the cell fate challenges are suppressed. Their results suggest that starvation has an impact on Notch signaling to stabilize cell fates.

Overall, this is a very good study revealing important aspects of cell fate regulation and implication of different signaling pathways in combination with the metabolic state of an organism. The experiments have been performed accurately and the conclusions are plausible. I support publication of this manuscript.

We would like to thank reviewer #3 for his positive and encouraging comments.

Minor comments:

The authors should consider using the term 'transdifferentiation' more carefully. It is being used several times in the abstract. However, it seems that there are in fact no real, or at least stable, transdifferentiation events. The gain of additional muscle marker-positive cells is due to proliferation and the anal sphincter cell shows only transiently the muscle gene expression. I'd suggest at least using the term as 'transient transdifferentiation' or similar.

Throughout the text, we have modified the wording by replacing «transdifferentiation» by «cell fate challenge».

With regard to the effects of the endoderm inducer END-1, it would be good to know if the authors tested whether there are also additional cells showing an intestinal fate reporters (e.g. elt-2::gfp).

We scored the number of cells expressing *pha-4::mCherry* upon ectopic HLH-1 or END-1 expression in *mes-2* animals. We observe that upon HLH-1^{ect}, the number of cells expressing *pha-4::mCherry* becomes highly variable, but in average remains centered around 19 cells as for wild-type animals. In contrast, upon END-1^{ect} expression, this number significantly increases. These data have been integrated as Figure S4.

December 17, 2018

RE: Life Science Alliance Manuscript #LSA-2018-00170-TR

Dr. Peter Meister
Institute of Cell Biology
University of Bern Baltzerstrasse 4
Bern CH-3012
Switzerland

Dear Dr. Meister,

Thank you for submitting your revised manuscript entitled "Polycomb and Notch signaling regulate cell proliferation potential during *C. elegans* life cycle". We appreciate the introduced changes and we would be happy to publish your paper in Life Science Alliance pending final revisions necessary to meet our formatting guidelines.

- please upload your manuscript text as a docx file
- please add callouts in the text to Fig4C, Fig5A

A. FINAL FILES:

-- High-resolution figure, supplementary figure and video files uploaded as individual files: See our detailed guidelines for preparing your production-ready images, <http://life-science-alliance.org/authorguide>

B. MANUSCRIPT ORGANIZATION AND FORMATTING:

Full guidelines are available on our Instructions for Authors page, <http://life-science->

alliance.org/authorguide

Thank you for your attention to these final processing requirements.

Sincerely,

December 18, 2018

RE: Life Science Alliance Manuscript #LSA-2018-00170-TRR

Dr. Peter Meister
Institute of Cell Biology
University of Bern Baltzerstrasse 4
Bern CH-3012
Switzerland

Dear Dr. Meister,

Thank you for submitting your Research Article entitled "Polycomb and Notch signaling regulate cell proliferation potential during *C. elegans* life cycle". It is a pleasure to let you know that your manuscript is now accepted for publication in Life Science Alliance. Congratulations on this interesting work.

DISTRIBUTION OF MATERIALS:

Again, congratulations on a very nice paper. I hope you found the review process to be constructive and are pleased with how the manuscript was handled editorially. We look forward to future exciting submissions from your lab.

Sincerely,
